# Single-cell profiling of *Anopheles gambiae* spermatogenesis defines the onset of meiotic silencing and premeiotic overexpression of the X chromosome

Nicole Page [1,5], Chrysanthi Taxiarchi [2,5], Daniel Tonge[3], Jasmina Kuburic[3], Emily Chesters [3], Antonios Kriezis[2], Kyros Kyrou[2], Laurence Game[4], Tony Nolan [1,6✉] & Roberto Galizi [3,6✉]

Understanding development and genetic regulation in the *Anopheles gambiae* germline is essential to engineer effective genetic control strategies targeting this malaria mosquito vector. These include targeting the germline to induce sterility or using regulatory sequences to drive transgene expression for applications such as gene drive. However, only very few germline-specific regulatory elements have been characterised with the majority showing leaky expression. This has been shown to considerably reduce the efficiency of current genetic control strategies, which rely on regulatory elements with more tightly restricted spatial and/or temporal expression. Meiotic silencing of the sex chromosomes limits the flexibility of transgene expression to develop effective sex-linked genetic control strategies. Here, we build on our previous study, dissecting gametogenesis into four distinct cell populations, using single-cell RNA sequencing to define eight distinct cell clusters and associated germline cell–types using available marker genes. We reveal overexpression of X-linked genes in a distinct cluster of pre-meiotic cells and document the onset of meiotic silencing of the X chromosome in a subcluster of cells in the latter stages of spermatogenesis. This study provides a comprehensive dataset, characterising the expression of distinct cell types through spermatogenesis and widening the toolkit for genetic control of malaria mosquitoes.

[1] Department of Vector Biology, Liverpool School of Tropical Medicine, Liverpool, UK. [2] Department of Life Sciences, Imperial College London, London, UK. [3] Centre for Applied Entomology and Parasitology, School of Life Sciences, Keele University, Keele, UK. [4] Genomics Facility, MRC London Institute of Medical Sciences, Imperial College London, London, United Kingdom. [5]These authors contributed equally: Nicole Page, Chrysanthi Taxiarchi. [6]These authors jointly supervised this work: Tony Nolan, Roberto Galizi. ✉email: tony.nolan@lstmed.ac.uk; r.galizi@outlook.com

Spermatogenesis is the process by which highly specialised sperm cells are created in the male gonad (Fig. 1a). In *Anopheles gambiae*, knowledge of this process is largely derived from the model organism *Drosophila melanogaster*. The stem cell niche at the apical hub of the testes consists of 3 cell types: the somatic hub cells and somatic stem cells, which offer support to germline development, and the germline stem cells (GSCs). The GSCs undergo repeated mitotic division to give rise to two daughter cells; one of which remains a stem cell anchored to the hub and the other begins differentiation into primary spermatogonium which is then displaced from the stem cell niche. Spermatogonia undergo multiple rounds of mitotic divisions to generate spermatocytes which enter meiosis. The resulting haploid spermatids differentiate into spermatozoa which constitute the mature male gametes[1].

The study of the mosquito germline has provided valuable tools for genetic vector control strategies. One such application is the use of germline regulatory elements for the expression of transgenes, with desired activation at different stages according to the control strategy. For example, homing-based gene drives rely on premeiotic activity of nucleases in diploid germ cells to effectively copy and paste the transgene into homologous chromosomes prior to meiotic division[2–6]. In other cases, nucleases designed to cleave the X chromosome require male-specific expression in haploid germ cells to shred X-bearing sperm and create a male-bias sex distortion in the progeny[7–9]. One major limitation of these strategies thus far has been the reduced efficiency caused by transgenic nuclease activity being insufficiently restricted to the male or female germline[10–13]. This is thought to be caused by leaky expression in the somatic cells and/or germline-expressed proteins being deposited in the zygote post-fertilisation. Consequently, the choice of promoter for transgene expression has a substantial impact on the efficiency of current control strategies[14], creating a need for genes, and corresponding regulatory elements, displaying more tightly regulated germline expression. Additionally, the development of other control strategies targeting male fertility[15,16] would be benefited from a wider range of male germline-specific genes being identified in mosquitoes.

During spermatogenesis, regulation of gene expression differs between the sex chromosomes and the autosomal chromosomes, primarily due to two mechanisms known as dosage compensation and Meiotic Sex Chromosome Inactivation (MSCI). Dosage compensation is a widely described cellular mechanism that balances the expression of sex-linked and autosomal genes in the heterogametic sex such as *A. gambiae* male mosquitoes, which carry only one copy of the X chromosome in each diploid cell, whilst females carry two X chromosomes. Previously, dosage compensation has been shown to be present in *A. gambiae* pupae with the exception of the male germline[17,18]. Additionally, it has been demonstrated that during the meiotic stage of *A. gambiae* spermatogenesis, genes on the X chromosome are repressed by a process resembling MSCI[17]. This meiotic silencing corresponds with previous studies showing transgenes inserted on the X chromosome[9,19]. MSCI is a process described in mammals that causes a widespread transcriptional silencing of the X chromosome during meiosis, after synapsis is complete, and persists throughout the rest of the germline[20,21]. The mechanisms, and exact start point, for this apparent MSCI in *A. gambiae* are largely unknown. A better understanding of this process could help identify a more precise point at which sex-linked genes are silenced and provide more flexibility in future transgene expression.

In this study, we aim to utilize the increased resolution of single-cell transcriptomics to further differentiate the male germline into distinct cell types. This provides a means to uncover genes with small windows of expression that may be hidden in larger cell populations and, consequently, reveal new

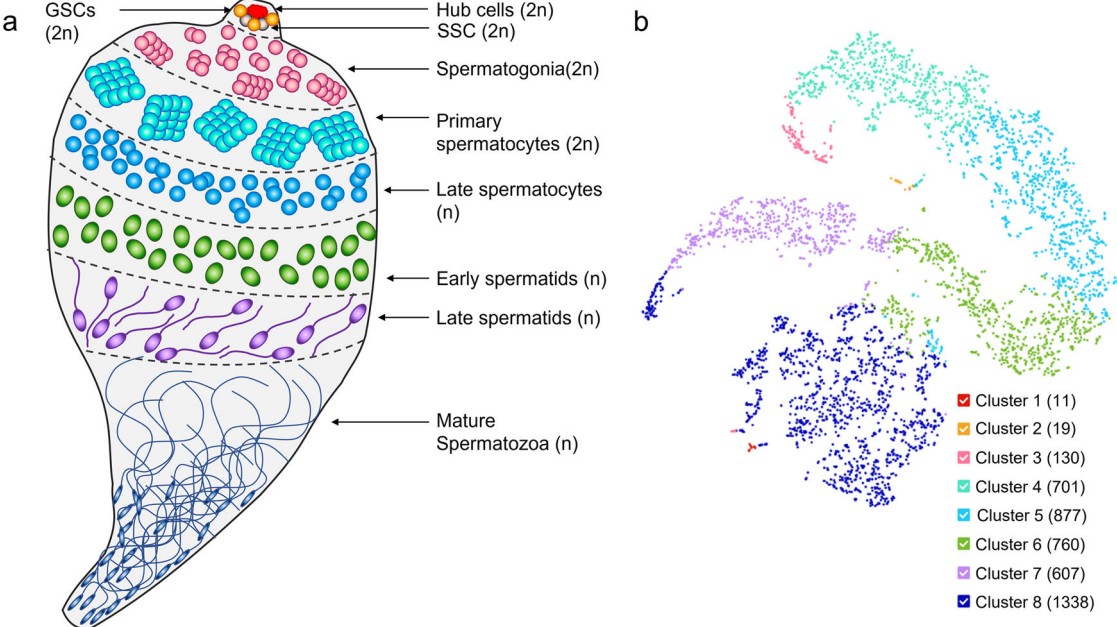

**Fig. 1 A single-cell population of the *Anopheles* male germline. a** Schematic of the different cell types of the male germline and their progression through spermatogenesis from diploid (2n) to haploid (n) cells. In the apical region of the testes the hub cells and somatic stem cells (SSC), together with the germline stem cells (GSCs), form the stem cell niche. GSCs divide to give rise to two daughter cells: one remaining a stem cell and the other developing into a primary spermatogonia cell. Spermatogonia undergo rounds of mitosis to produce primary spermatocytes. Meiotic division then gives rise to haploid spermatocytes which eventually develop into maturing spermatids and spermatozoa. **b** t-SNE dimensionality reduction plot of single-cell RNA sequencing dataset from *A. gambiae* testes. Each dot represents a cell which is mapped according to transcriptional similarities with neighbouring cells. Cells are colour coded by K-means cluster with the number of cells in each cluster shown in parentheses.

gene targets that are important for spermatogenesis as well as provide much needed alternatives to current germline promoters used for the genetic control of mosquito vectors.

## Results and discussion

**Isolation of single germline cells from the *Anopheles* male gonads for single-cell RNA sequencing.** We used transgenic mosquito strains carrying germline-specific fluorescent markers to isolate the germ cells from the testes of *A. gambiae* adult males. Individuals expressing GFP and mCherry, under the *vasa2* and ß2-tubulin promoters, respectively, allowed us to distinguish germline cells from the somatic tissues of dissected male gonads[17]. Fluorescently labelled cells from these mosquito lines were isolated from adult testes and pooled to create a suspension of male germline-specific single-cells. During the fluorescent screening process, we also filtered out a large proportion of the later germline cells (determined by smaller cells that expressed mCherry) to prevent the high proportion of mature spermatids and spermatozoa in the testes from dominating the sample (Supplementary Figure 1). Single-cell 3′ RNA sequencing with 10× genomics provided transcriptomic data for 4443 individual cells with a median of 2849 genes per cell and a mean of 56,907 reads per cell. In total 88.7% of reads mapped to the genome with 65.2% mapped confidently to exonic regions. Cells with a minimum expression of 200 genes and genes expressed in at least one cell were included for downstream analysis.

**Unsupervised spatiotemporal progression and differential expression analysis of single cell clusters elucidates cell-type annotation for mosquito spermatogenesis.** We used k-means clustering to segregate the dataset into a predefined number of clusters varying from 2 to 10. We chose to characterise 8 distinct cell clusters for downstream analysis with the smallest cluster of 11 cells and the largest with 1338 cells (Fig. 1b). Eight was the maximum number of clusters that could be ordered in a meaningful pseudotime trajectory analysis and annotated with available germline marker genes to assign the corresponding inferred cell-type (Fig. 2a).

A pseudotemporal trajectory analysis was performed to calculate the predicted distance of differentiation between cells and infer this trajectory as pseudotime values[22–24] (Fig. 2b; Supplementary Fig. 2). We used the cells showing the highest expression of the premeiotic *vasa* gene (Fig. 2c) to determine the root node of the trajectory, where pseudotime is equal to zero, as this is indicative of premeiotic germ cells at the start of spermatogenesis[25]. The remaining cells were automatically assigned pseudotime values based on the predicted level of differentiation from cells at the root node. This allowed us to determine the likely order of the cell clusters in terms of stages of spermatogenesis. Two of the smaller clusters, later characterised as the hub cells and GSCs, were not encapsulated by the trajectory path (Fig. 2b). We, therefore, relied on available marker genes and *Drosophila* orthologs to complete the cell-type annotation for all clusters.

A globally distinguishing differential expression analysis revealed genes that were significantly enriched (Log2FC > 1 and $p < 0.05$) in each cell cluster compared to all remaining cells (Supplementary Data 1). Using a combination of genes with known expression in the *A. gambiae* germline[17], and orthologs of *Drosophila* germ-cell markers[26], we were able to assign each cluster with an inferred germline cell-type based on enrichment of expression for these markers (Fig. 2c–l; Supplementary Figure 3). The premeiotic *vasa* gene (AGAP008578) was used to identify cells of the early germline and *β2-tubulin* (AGAP008622) expression helped characterise cells from the

meiotic stage onwards. The somatic hub cells were characterised by a significant enrichment for AGAP029564 (Log2FC: 9.98, $p = 4.90E\text{-}11$), an ortholog of the *Drosophila* hub-cell marker *Fas3*[26]. These cells also show a low level of expression from the mCherry fluorescent marker, preventing them from being removed during germline cell filtering via Fluorescence Activated Cell Sorting (FACS). A significant enrichment for expression of *geminin* (AGAP000496) (Log2FC: 2.71, $p = 0.002$) and *squid* (AGAP000399) (Log2FC: 3.96, $p = 2.47E\text{-}07$), genes involved in the maintenance of GSCs[27–30], was used to define the GSC cell cluster. Enrichment for *geminin* (Log2FC: 3.20, $p = 7.81E\text{-}34$) and *squid* (Log2FC: 3.02, $p = 1.50E\text{-}21$) expression also persisted into the primary spermatogonia cells. The primary spermatogonia were characterised by the additional expression of *Aurora kinase* (AGAP007855) which was significantly enriched in this cell cluster (Log2FC: 1.98, $p = 7.93E\text{-}12$). Enrichment for *Aurora kinase* was also observed in the following cell cluster (Log2FC: 1.67, $p = 2.35E\text{-}31$) which was defined as late spermatogonia/primary spermatocytes. *Aurora kinase* is an ortholog of the *Drosophila Aurora A* and *Aurora B* genes that are important for mitotic spindle formation and recruiting factors for chromatin condensation[31–34]. *Aurora B* has also been linked to meiotic chromatid cohesion[35], supporting the characterisation of these cell clusters as mitotic spermatogonia and meiotic primary spermatocytes. The late spermatogonia/primary spermatocytes cell cluster also exhibited significant expression of *argonaute 3* (AGAP008862) (Log2FC: 1.01, $p = 2.80E\text{-}11$) and *aubergine* (AGAP011204) (Log2FC: 1.18, $p = 1.23E\text{-}16$). *Aubergine* is part of the *argonaut* family in *Drosophila* and is involved in gene silencing in the germline[36] indicating a potential involvement in MSCI starting in spermatocyte cells. Additionally, the *RacGAP1/tumbleweed* gene (AGAP008912), expressed in the ring canals of premeiotic germ cells to permit cytoplasmic sharing[37,38], showed enrichment in the GSCs (Log2FC: 2.83, $p = 0.005$), Primary spermatogonia (Log2FC: 2.61, $p = 1.12E\text{-}15$) and Late spermatogonia/Primary spermatocyte (Log2FC: 1.90, $p = 1.97E\text{-}26$) cell clusters, confirming their premeiotic cell state.

Expression patterns of *ß2-tubulin* (AGAP008622), expressed from meiosis onwards[17], and *elf4E-5* (AGAP007172), involved in post-meiotic translation initiation[39], confirmed the identification of the later stages of spermatogenesis, namely the spermatocytes and early spermatids. During the final stages of sperm development, DNA histones are replaced with protamines to facilitate chromatin hypercondensation and the resultant transcriptional silencing observed in mature sperm cells[40–45]. The *A. gambiae* protamine gene (AGAP028569) showed significant enrichment (Log2FC: 1.01, $p = 2.92E\text{-}05$) in what was determined to be the late spermatids. We were also able to characterise the mature spermatozoa based on a significant enrichment for the ortholog of the *Drosophila* axonemal dynein protein *Dnah3* (AGAP007675) (Log2FC: 1.44, $p = 0.007$) which is critical for sperm motility[46].

Additionally, we investigated whether the most significantly enriched genes per cell cluster (Supplementary Figure 4) corresponded to our inferred cell-type assignment. The top enriched genes in the hub cells and GSCs display highly specific expression to their associated cell cluster (Supplementary Fig. 4a, b) compared to the later cell clusters (Supplementary Fig. 4c–h). This continuity in gene expression spanning different spermatogenic stages explains why, without a more substantial catalogue of marker genes in *A. gambiae*, it is difficult to state the exact point in which spermatogonia develop into spermatocytes and, later, spermatids. Although many of these top enriched genes have unknown function, a few can add support to the cell-type definitions. One of

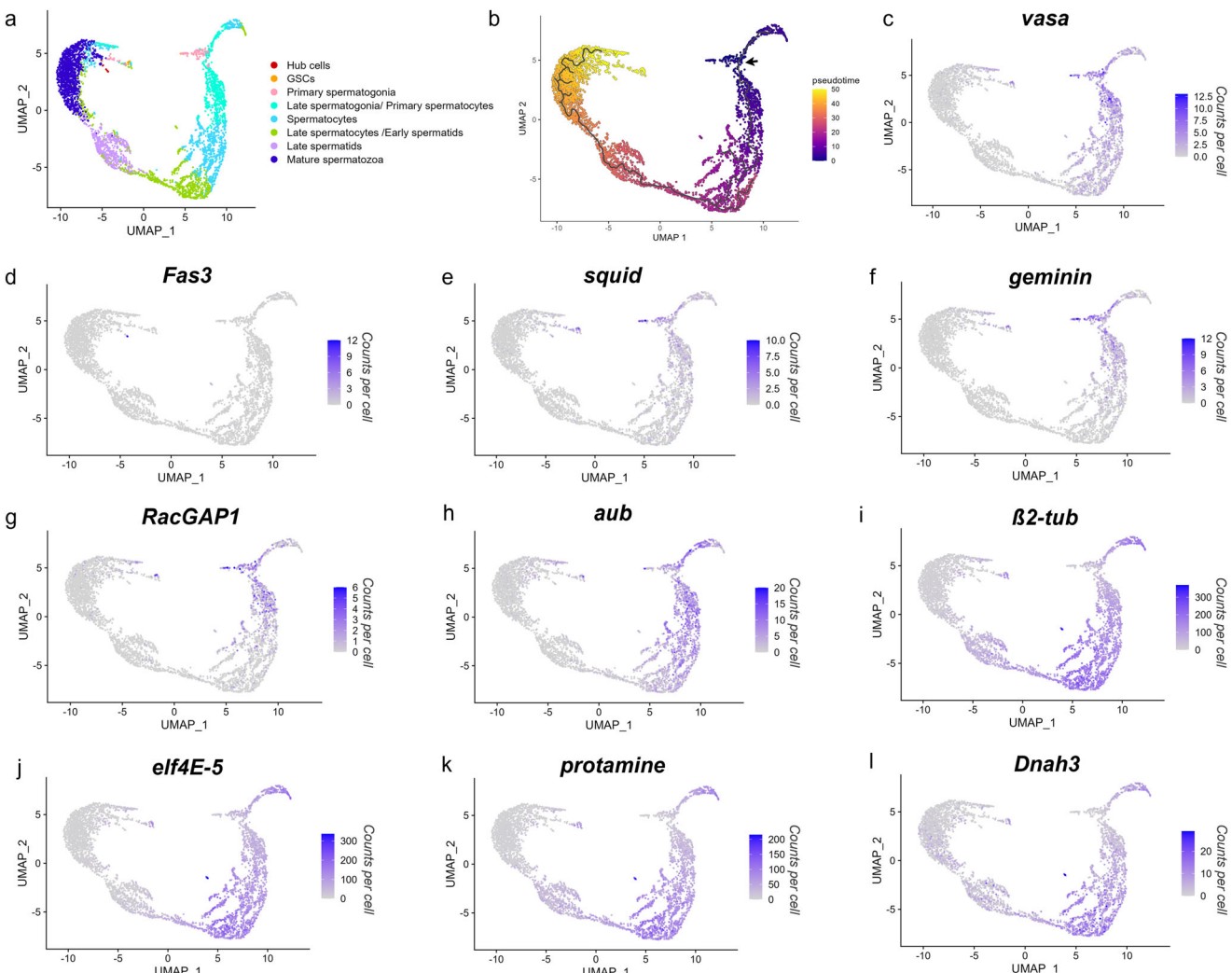

**Fig. 2 Pseudotime trajectory of single cells and expression profiles of annotated germline marker genes. a** UMAP (Uniform Manifold Approximation and Projection) non-linear dimension reduction is used to plot single cells (single dots) in relation to each other in 2D space. **b** Pseudotime trajectory plot showing predicted differentiation of germline cells (black line). The root node at which pseudotime is zero (indigo) was selected using known expression of the *vasa* gene (AGAP008578) in the premeiotic germline cells (black arrow). Pseudotime values are automatically assigned to remaining cells with higher values (yellow) representing a further differentiation from cells near the root node. **c–l** Linear expression levels of marker genes (counts per cell) for *vasa* (AGAP008578), *Fas3* (AGAP029564), *squid* (AGAP000399), *geminin* (AGAP000496), *RacGAP1* (AGAP008912), *aubergine* (AGAP011204), *ß2-tubulin* (AGAP008622), *elf4E-5* (AGAP007172), *protamine* (AGAP028569) and *Dnah3* (AGAP007675) are displayed on a graded colour scale.

the top enriched genes in the hub cells, AGAP000472, is the ortholog of Drosophila *short gastrulation* (*sog*), which is involved in maintaining the stem-cell niche in the hub via the Bmp signalling pathway[47,48]. Interestingly, the top 3 enriched genes for mature spermatozoa (AGAP028393, AGAP028391, AGAP028370) are mitochondrial genes, likely supporting the energy required for sperm motility.

**Cell cluster enriched genes show predicted function compatible with spermatogenic stages.** A gene ontology analysis of cluster enriched genes was used to reveal significant overrepresentation of biological processes and/or molecular function in these cells (Supplementary Data 2). The GSCs show a statistically significant over-representation of cell signalling/communication and organelle organisation. For organelle organisation, genes include the *A. gambiae* orthologs for *Rho kinase* (AGAP000406) and *Myosin VI/ Jaguar*, which regulate stem cell-like asymmetric cell division in *Drosophila*[49–51], as well as *Moesin* (AGAP000562) which acts

specifically in the germline stem cell niche[52]. The primary spermatogonia exhibited an over-representation for biological processes such as, RNA Processing, transcription, translation, RNA splicing, RNA modification, ribonucleoprotein complex assembly and gene expression. This is characteristic of the high levels of expression expected in these cells which are undergoing rapid growth and mitotic division. The primary spermatogonia also exhibited the highest specificity in gene expression, with the largest number of genes exclusively enriched in this cell cluster (Fig. 3a), supporting high levels of specific expression. The late spermatogonia/primary spermatocytes cell cluster is also enriched for RNA splicing, translation and gene expression processes. Additionally, this cell cluster also displays an overrepresentation for spermatogenesis, male gamete generation, negative regulation of transcription and negative regulation of RNA biosynthetic process. This is indicative of the meiotic silencing observed during the meiotic phase of spermatogenesis[17]. Alongside the enrichment for expression of *aubergine* in these cells, this supports our hypothesis that the late spermatogonia/primary

spermatocyte cell cluster comprises the onset of MSCI. Genes enriched in the late spermatocytes/early spermatids display trends suggestive of general transcriptional silencing such as a significant underrepresentation of gene expression, RNA metabolic process and nucleic acid metabolic process. This correlates with the transcriptional inactivity observed in early spermatids in *Drosophila*[53,54]. Finally in the mature spermatozoa, we see an overrepresentation of biological processes including aerobic respiration, cellular respiration, energy derivation by oxidation of organic compounds and generation of precursor metabolites and energy as well as molecular functions microtubule motor activity and cytoskeletal motor activity. This high level of respiration and motor activity is indicative of the energy requirements and mechanical functions for effective sperm motility. This is further supported by an enrichment of mitochondrial DNA genes in this cluster (Fig. 3b, Supplementary Data 1, Supplementary Data 3).

**Chromosome-wide expression analysis indicates overexpression of the X-chromosome in GSCs and the onset of MSCI.** We calculated the overall RNA content per cell (total number of RNA molecules) that mapped to either the

autosomes (Fig. 4a), X chromosome (Fig. 4b) and Y contigs (Fig. 4c) as an indicator for the level of expression in each cluster (Supplementary Data 4). RNA levels from the autosomal chromosomes steadily increase throughout the early germline, peaking in the late spermatocytes, before declining in the post-meiotic maturing spermatids. Conversely, expression from the X chromosome peaks in the GSCs before a rapid reduction in expression throughout the following stages of spermatogenesis. This shows an apparent increased expression of genes on the X chromosome in the GSCs, with 98% (529/ 541) of significantly enriched genes being located on the X chromosome (Fig. 3b), drastically more than what would be expected by a random distribution of chromosome location. This suggests a possible selection for genes important at this developmental stage to be located on the X. It was previously hypothesised that the X chromosome could be enriched for genes important in early spermatogenesis due to a potential selective pressure for recessive alleles, owing to males having just one copy of X-linked genes, that are beneficial to the male germline[55,56]. Other studies have also found an abundance of X-linked genes in premeiotic spermatogonia cells, indicating a similar phenomenon in mammalian premeiotic germ cells[57].

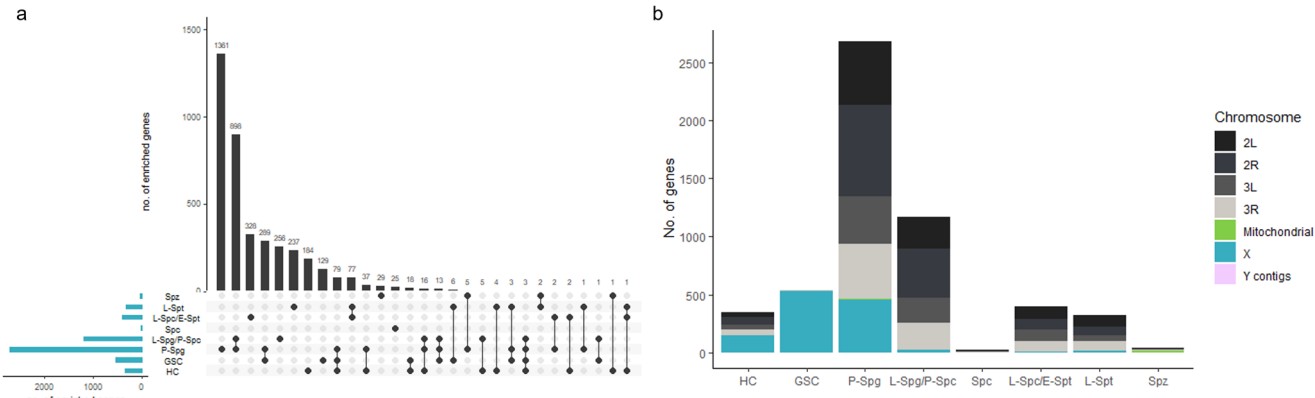

**Fig. 3 Distribution of enriched genes across germline cell clusters.** The number of genes significantly enriched (LogFC >1; *p* < 0.05) for each cell cluster; Hub cells (HC), Germline stem cells (GSC), Primary spermatogonia (P-Spg), Late Spermatogonia/Primary spermatocytes (L- Spg/P-Spc), Spermatocytes (Spc), Late spermatocytes/Early spermatids (L-Spc/E-Spt), Late spermatids (L-Spt) and Mature spermatozoa (Spz). **a** UpSet plot showing the total number of enriched genes per cluster (horizontal blue bars) and the number of enriched genes that are unique or shared between clusters (vertical black bars). The clusters that are enriched for each gene set are shown by black dots running vertically below each bar. **b** Breakdown of cluster enriched genes by chromosomal location.

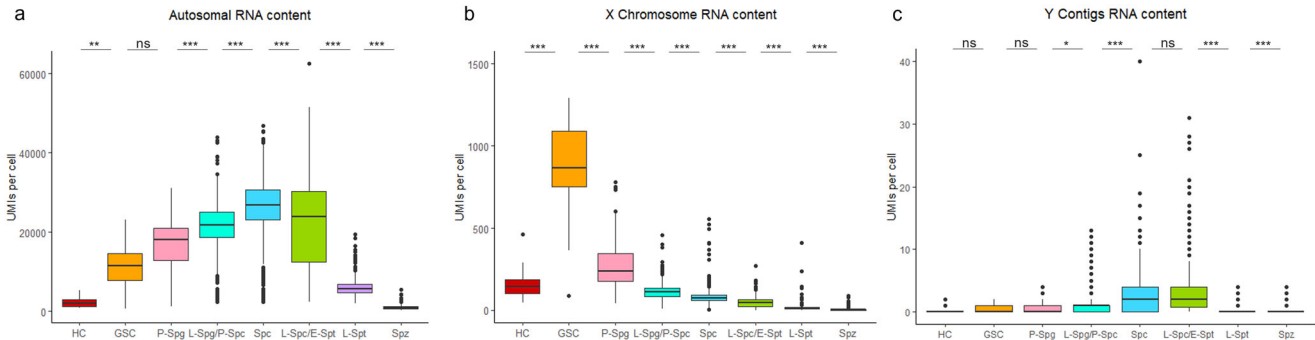

**Fig. 4 Total RNA content of Autosomal, X-linked and Y-linked genes.** RNA content is calculated by taking the sum of all unique molecular identifiers (UMIs), which correspond to a single RNA molecule, that map to autosomal **a**, X chromosome **b**, or Y contigs **c** for each cell. Hub cells (HC), Germline stem cells (GSC), Primary spermatogonia (P-Spg), Late Spermatogonia/Primary spermatocytes (L- Spg/P-Spc), Spermatocytes (Spc), Late spermatocytes/Early spermatids (L-Spc/E-Spt), Late spermatids (L-Spt) and Mature spermatozoa (Spz). Statistical differences calculated using Wilcoxon tests with adjusted *p*-values using the Bonferroni correction for multiple testing (*p*.adj <0.0001 ***, *p*.adj <0.001 **, *p*.adj <0.05 *). The lower and upper quartiles are defined by the box with the median represented within. Whiskers show the maximum and minimum range of the data with any outliers beyond this displayed as circles.

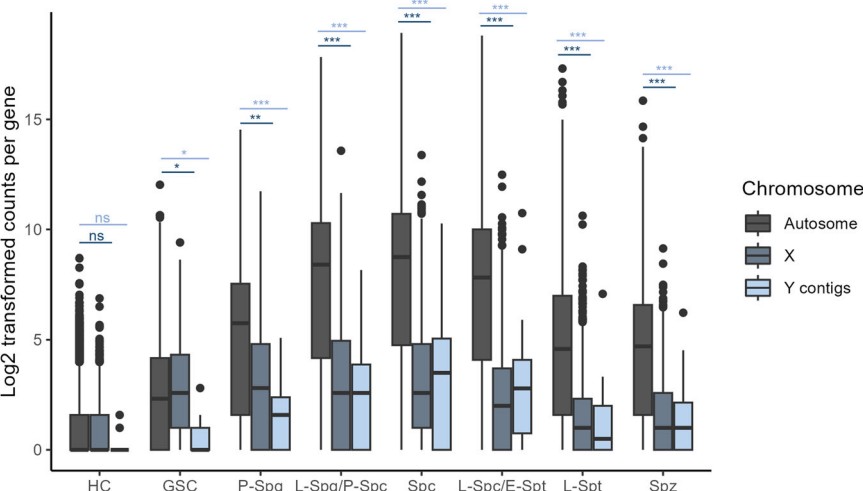

**Fig. 5 Comparison in expression between genes located on the sex and autosomal chromosomes.** Log2(counts +1) per gene are displayed for all genes on the X chromosome or Y contigs compared to all genes located on an autosome (2 R, 2 L, 3 R or 3 L) for each cell cluster. Hub cells (HC), Germline stem cells (GSC), Primary spermatogonia (P-Spg), Late Spermatogonia/Primary spermatocytes (L-Spg/P-Spc), Spermatocytes (Spc), Late spermatocytes/Early spermatids (L-Spc/E-Spt), Late spermatids (L-Spt) and Mature spermatozoa (Spz). Wilcoxon tests were used to determine a significant difference between autosomal and X-linked or Y-linked gene counts (p.adj <0.0001 ***, p.adj <0.001 **, p.adj <0.05 *). Adjusted p-values were calculated using the Bonferroni correction for multiple testing. The lower and upper quartiles are defined by the box with the median represented within. Whiskers show the maximum and minimum range of the data with any outliers beyond this displayed as circles.

Genes mapping to the Y contigs generally show very low levels of expression throughout spermatogenesis with a slight increase in spermatocytes and spermatids. Some of the Y contigs have been previously described as having multiple copies, some of which on the autosomes[17,58], that could explain a slight meiotic trend in expression (Fig. 4c).

We also sought to characterise the presence of dosage compensation in the *A. gambiae* male germline by comparing the expression of genes on the X and autosomal chromosomes within each cluster (Fig. 5, Supplementary Data 5). In the presence of dosage compensation in males, carrying only one X chromosome, the level of expression from a single copy of X-linked genes is expected to be compensated to match the expression observed in females carrying two copies. We compared the median expression values of genes located on the X vs autosomal chromosomes to establish if the level of X-linked gene expression is compensated to match that seen in the two copies of autosomal genes. Consequently, an X/A ratio of ~1 would be indicative of active dosage compensation while a ratio of ~0.5 is expected in cells lacking dosage compensation due to X-linked genes having half the copy number of those on the autosome. The somatic hub cells did not show a significant difference in expression between the X and autosomal chromosomes, indicating the presence of dosage compensation in these cells (Fig. 5; Supplementary Table 1). Interestingly, the GSCs exhibited a significantly higher expression of X-linked genes compared to the autosome (Wilcoxon test with Bonferroni correction; X/A = 1.113, p = 0.002), confirming the overexpression of the X chromosome in these cells. This demonstrates the increased resolution of single-cell transcriptomics in revealing the cell-type exhibiting an overexpression of the X chromosome that was previously hinted in previous bulk sequencing of premeiotic germ cells[17]. Similar studies in *Drosophila* also revealed a significant increase in expression from the X chromosome in GSCs that was attributed to potential overcompensation[26]. Our previous work also showed that the pre-meiotic germ cells displayed X/A ratios ranging from <0.5 and >1 depending on the threshold for gene expression, suggesting that a low number of X-linked genes

are overexpressed[17]. In the primary spermatogonia, an X/A ratio of significantly less than 1 (Wilcoxon test with Bonferroni correction; X/A = 0.488, p = 2.10E-71) suggests that the overexpression of the X chromosome does not extend to the spermatogonia. This is also indicative of an absence of dosage compensation, differing from what was previously observed in *Drosophila* where spermatogonia display equal expression from the X and autosomes[26]. The following germline cell clusters all exhibit X/A ratios lower than expected from a lack of dosage compensation alone (Supplementary Table 1). This suggests that silencing of the X is likely present in these cells demonstrating the onset of MSCI in the Late spermatogonia/ Primary spermatocyte cell cluster (Fig. 5; Supplementary Table 1). This correlates with previous evidence of MSCI being present in meiotic and post-meiotic germline cells and offers a more defined onset of silencing in the Late spermatogonia/ Primary spermatocyte cells[17]. Similar studies in *Drosophila* also observed reduction in X chromosome expression initiating in spermatocytes[26,59].

Performing the same analysis using transcript reads aligned to the Y contigs compared to the autosomes showed significantly lower expression in the Y contigs indicating a lack of dosage compensation in all germline cell clusters (Fig. 5). Therefore, the overexpression of the X chromosome seen in GSCs does not appear to extend to the genes on the Y.

**Conclusions**

This study provides a detailed transcriptomic profile of the individual cell types in the male germline of the main malaria vector *A. gambiae*. Although a comprehensive and rigorous identification of defined cell-type groups would require a broader repertoire of marker genes, which is currently lacking in mosquitoes, the benefits of using single-cell transcriptomic resolution were made apparent by the possibility to reconstruct the developmental trajectory of spermatogenesis using a combination of unsupervised analysis tools, characterised germline genes and *Drosophila* orthologs. This allowed us to pinpoint an overexpression of X-inked genes in a small population of cells, identifiable as GSCs, previously hidden in bulk cell population

analysis. Additionally, we were able to pinpoint the subgroup of germline cells where meiosis and meiotic silencing appears to commence. Further characterisation of the genetic markers of MSCI/DC, for example through in situ transcript hybridization, is required to further define the exact cell types in which the onset of these mechanisms occurs.

The increased resolution of single-cell transcriptomics allowed us to identify genes with small windows of expression that were previously undetectable via bulk sequencing of whole tissues and/ or larger cell populations. Having established the different stages of spermatogenesis within this dataset, the expression profiles from individual germline cells can be used as a resource for finding genes expressed during specific time points in the male germline. This dataset will provide valuable insight for future study of the *Anopheles* germline as well as aid in the design of genetic technologies utilising germline-specific processes and/or expression patterns.

## Methods

**Preparation of male germline cell suspension and single-cell RNA sequencing**. A total of 191 testes from one-day old mosquitoes expressing β2:mCherry (β2mC-2 line) and 194 expressing vasa:eGFP (YVasG line) were dissected and homogenised to produce a single cell suspension as previously described[17]. The cell suspension was analysed and sorted using a BD FACSDiva sorter (BD Biosciences) with a 100 μm nozzle and 28 PSI pressure. TO-PRO-3 was used to discriminate live from dead cells and forward scatter height (FSC-H) against area (FSC-A) was used for doublet exclusion in the sample during the FACS process. Germline cells were sorted based on the expression of mCherry and eGFP fluorescent markers and size. An equal number of small (P5 and P7) and large cells (P6 and P8) were selected to prevent mature sperm being over-represented in the final single-cell sample (Supplementary Figure 1). A total of 8600 cells were sorted and subsequently used for library preparation using the Chromium Single Cell 3′ v2 Reagent Kit (10× Genomics) following the manufacturer's instructions and sequenced using the HiSeq 2500 System (Illumina) with paired end reads of 100 nucleotides (Genomics Facility, MRC London Institute of Medical Sciences - UK). Raw reads were aligned to the VectorBase-59_AgambiaePEST reference genome with added Y contigs using the CellRanger pipeline (10× Genomics). Normalisation and unsupervised K-means clustering was also performed using CellRanger to create a cloupe file and associated cell count matrix with a total of 4443 cells for downstream analysis.

**Differential Expression and Gene ontology analysis**. A globally distinguishing differential expression analysis was performed in loupe browser to find cluster enriched genes by individually comparing each cluster, to all other cells in the dataset. Genes were considered enriched in a cluster if the log2 Fold Change in expression was >1 with a $p$-value of < 0.05; $p$-values were adjusted using the Benjamini-Hochberg correction for multiple testing[60]. Genes found to be significantly enriched were input for a gene ontology analysis using the PANTHER (Protein Annotation Through Evolutionary Relationship) classification system[61] to identify any significantly ($p < 0.05$) overrepresented pathways or molecular functions in each cluster.

**Pseudotime trajectory analysis**. We used Seurat single-cell analysis tool[62] to import our cell data, along with previously determined K-means clustering, into a Seurat object. UMAP coordinates and cluster information was then imported into Monocle 3[22–24] to perform a pseudotime trajectory analysis with the root node selected manually based on a visible peak in vasa expression. Pseudotime values were used to predict levels of differentiation in cell clusters based on divergence from the root node.

**MSCI and dosage compensation analysis**. A threshold was set for genes that were expressed in at least one cell cluster (at least one read detected) to be included in the analysis to remove the influence of genes that are not expressed in the testes. We transformed the counts for each gene by log2(counts +1) to determine 'log2 counts per gene' and used these values to determine the expression ratio between genes on the X chromosome, Y contigs and autosomal chromosomes. The median log-transformed counts were compared for each cluster individually using a Wilcoxon test, with Bonferroni multiple testing correction[63], to determine a significant variation from a 1:1 ratio that would be expected in the presence of dosage compensation. The total RNA content for X, Y contigs and autosomal chromosomes were also calculated using the UMI counts which represent the number of RNA molecules in each cell. We took the sum of UMI counts for each gene per cell and separated cells by cluster. Genes were then filtered for their chromosomal location. Multiple Wilcoxon tests with Bonferroni multiple testing correction were used to determine a significant increase/decrease in RNA content between consecutive cell clusters.

**Statistics and reproducibility**. In total 4443 individual cells were used in the downstream analysis for this study. Differential expression analysis was performed in loupe browser using the globally distinguishing analysis across all clusters; $p$-values were adjusted using the Benjamini-Hochberg correction for multiple testing[60]. Further analysis at the gene level was performed excluding genes that were expressed in at least one cell in the dataset. Non-parametric Wilcoxon tests were performed in R and used to compare both raw and log-transformed count data between chromosomes and cell clusters. Adjusted $p$-values for the Wilcoxon tests were calculated using the Bonferroni correction for multiple testing[63].

**Reporting summary**. Further information on research design is available in the Nature Portfolio Reporting Summary linked to this article.

## Data availability

The original sequencing data for this paper is available online under accession number PRJNA971569. Source data for Figs. 3–5 are provided in Supplementary Data 3–5.

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

## Acknowledgements

This study was supported by a research grant from the BBSRC (BB/W014661/1) and a global health research seed grant from the Wellcome Trust. We thank Ivan Andrew at the London Institute of Medical Sciences Genomics Facility for technical assistance during library preparation and sequencing. We thank the LMS/NIHR Imperial Biomedical Research Centre Flow Cytometry Facility for the support. The views expressed are those of the authors and not necessarily those of the NHS, the NIHR or the Department of Health. The salary of C.T. was supported by the Defense Advanced Research Projects Agency (HR0011-17-2-0042). The views, opinions and/or findings expressed should not be interpreted as representing the official views or policies of the Department of Defense or the U.S. Government. The salary of K.K. and A.K. were supported by the Bill & Melinda Gates Foundation (OPP1210755).

## Author contributions

N.P., C.T., T.N., and R.G. conceived the project and designed the research; N.P., C.T., D.T., J.K., E.C., A.K., K.K., and L.G. performed the experiments and/or bioinformatic analysis; N.P., C.T., T.N., and R.G. wrote the paper with input from all authors.

## Competing interests

The authors declare no competing interests.
