## [Peer Review File · Communications Biology]

Reviewers' comments:

Reviewer #1 (Remarks to the Author):

This study by Page et al. uses RNA expression profiling of testes single cells to analyze spermatogenesis cell types in the mosquito *Anopheles gambiae*, which is an important vector of human diseases. One of the main goals of this study is to develop a catalogue of candidate genes with stage-specific expression during spermatogenesis, which can then be used for fine tuned activation of transgenes for various vector control strategies. Another aim of this study is to identify the precise stage at which meiotic sex chromosome inactivation (MSCI) is initiated. In addition to the general interest in the mechanisms of MSCI, understanding the stage at which it starts can facilitate candidate selection to avoid issues with artificially silencing X-linked genes.

The study is executed very well and the analysis clear and succinct. I think it is very clever to use the gene ontology results on the cell clusters and the expression pattern of *aub* to infer the onset of MSCI, and I think the resource will indeed be valuable for the community. One main suggestion: I thought the authors could have done a bit more with the pseudotime trends to corroborate the enriched cell cluster groupings. For example, identify genes that directly correlate with the pseudotime trajectories in addition to the cluster enrichment and DE between cell clusters. It is not guaranteed that this analysis would yield additional candidates for each cell type, but the authors should at least consider it.

My other suggestions below are fairly minor.

- Line 90-91: There's a strange reference error here.
- Figure 3: It would be preferable to plot the normalized log-transformed expression values to highlight abundance patterns more clearly. (Based on the value ranges, I'm assuming these are not log-transformed).
- Figure 5: legend typo "...of from..."
- Line 234-241: formatting issue.

Reviewer #2 (Remarks to the Author):

Summary: Page et al. present work to analyze the transcriptome of *Anopheles gambiae* testes. This organ has been previously studied by the authors at the cell compartment level using FACS using these same two transgenic reporters followed by high-throughput RNAseq sequencing. Their analysis has been repeated in more detail by single cell RNAseq. While incremental, this technology is better suited to describe the progression of spermatogenesis and capture lowly represented cell populations. While X-linked silencing was known, the authors identify overexpression of X-linked genes in germline stem cells, a tiny fraction of cells. As well as biological interest, this analysis may be of benefit to the wider community to identify regulatory elements for transgene expression that are specific for certain stages of spermatogenesis, although the authors do not validate any new regulatory elements, nor expression patterns by in situ hybridization.

Overall comments:

The data are convincing, and the paper is well written. I have mostly minor comments for clarity. The bioinformatics analysis is interesting and relevant to identifying the onset of germline silencing.

There seems to be no reporting of results on the other sex chromosome, the Y chromosome, despite mapping to Y contigs. Is this to be published elsewhere? If not, some comments on the findings would be appropriate e.g. Is Y chromosome also silenced during meiosis and post-meiosis? Are Y-bearing haploid cells transcriptionally different to X-bearing haploid cells?

The power of scRNAseq is in its ability to examine heterogeneity in cell populations but the authors do not specifically examine the most/least variable genes in each cluster. This would seem like a missed opportunity given this dataset.

Detailed comments:

Abstract

No comments.

Introduction

'...transgene expression being insufficiently restricted to the germline. This is thought to be caused by leaky expression in the somatic cells and/or germline-expressed proteins being deposited in the zygote post-fertilization.'

This point seems to be applicable to other strategies as well, not just those where sperm transfers proteins into the zygote but also maternal deposition into the egg is just as relevant. Since eggs and sperm are within the germline, I would say this is a protein/transcript stability issue rather than a lack of restriction to the germline.

'Dosage compensation is a widely described cellular mechanisms that equalizes the expression of X-linked genes in organisms containing a sex-determined number of X chromosomes.' This sentence is badly worded in my view and should be revised as it is a very broad statement and the organism is not specified. Sex is not necessarily determined by the number of X chromosomes but by the ratio of X to Autosomes or the presence of a Y chromosome e.g. XY and XXY are both human males.

Results and Discussion:

Isolation of single germline cells from the Anopheles male gonads for single-cell RNA sequencing

Additional details of the scRNAseq results would be helpful if presented, particularly relating to cell quality control, for example, cut-off values for reads per cell, total number of reads, mean reads per cell, % mapping to genome.

Typo: ... trajectory analysis (REF error). was performed...

Figure 1b is not referred to in the main text

Unsupervised spatiotemporal progression of single-cell clusters elucidates expression of cell-type enriched genes throughout mosquito spermatogenesis

The clusters described in Figure 3 and the text differ to those represented in Figure 1. The same region of the t-SNE plot is labelled as Cluster 4 (same color as primary spermatocytes) in Figure 1 and as Spermatogonia / Primary spermatocytes in Figure 3 and I find this discrepancy confusing. Why haven't spermatogonia been distinguished from primary spermatocytes? Consider adding some blue color to the primary spermatocytes region in Figure 1b.

Dnah3 doesn't appear to be particularly enriched in spermatozoa in the t-SNE plot as many cells do not express it. Are there other genes that might be more convincing?

How does expression of the fluorescent marker proteins compare with the endogenous gene expression of vasa and B2 tubulin?

Differential expression analysis highlights genes with predicted functions compatible with apparent spermatogenic stages – Typo in title?

In the cases of cluster 1 and 2, and 5 and 8, there are large differences in the number of cells being tested against 'the rest'. Are the gene enrichment results similar if fewer cells are randomly sub-sampled?

Are adjusted p-values presented in Table S1? What is the correction for multiple testing? Consider adding a statistical section to the methods.

Typo:...such as RNA processing, ...

Chromosome-wide expression analysis indicated overexpression of the X-chromosome in GSCs and the onset of MSCI

It's kinda odd to derive a ratio of 1 from median expression values of 0/0 for Hub cells when Figure 5 shows RNA content derived from both chromosome locations. What about if you restrict to genes detected in the cluster?

'This suggests that additional silencing...' Why is this additional? There is a lack of dosage compensation and a lack of silencing in primary spermatogonia.

Table 1: To claim silencing, it would seem appropriate to also test for deviation from a ratio of 0.5.

Typos:

... shows an apparent increase ...

... distribution of chromosomal location. This ...

Conclusions

Typo:

... an overexpression of X-linked genes...

Methods

How many biological replicates were done?

Please specify the repository where this data will be publicly available.

Figures

Figure 1b is not referred to in the main text.

Some panels of Figure 3 are not referred to in the main text.

Figure 5 title has a typo: ... content of from autosomal...

Figure 6: 1-star significance is not defined in the legend. Wilcoxon should have uppercase W

The headings Client text box are not very intuitive in Table S2

Reviewer #3 (Remarks to the Author):

What happens to sex chromosome expression during spermatogenesis and the prospects of using these properties to engineer vector control lines is topical and interesting. The title of the manuscript

coveys a deep analysis of the topic. We see a hint of this in the very interesting figure 5. Unfortunately, the manuscript is thin, probably premature, and does not really deliver strong conclusions. It will take some work, but I hope the authors put in the time rather than just turning around the manuscript, as the importance of a convincing work would be very high.

1) While annotating cell types is a long a tedious task, it is of critical importance. The literature is beginning to be polluted with poor annotations, that are then propagated. I find many of the elements of fig 3 worrisome, for example, the authors show genes that I would expect to see late, e.g. protamines and beta tubulin, expressed in the spermatogonia. The hub annotation is not at all convincing. Some of this might be due to the graded scale, which made it hard for me to distinguish high and low expression – so I might be overly harsh as a result. I think the authors should reexamine the clustering granularity and try to see if other resolutions and parameter adjustments could help make the stage separations better. I am not asking for distinct clusters, as this is a continuum, but I suspect this could be improved. If not, I don't think it's a fatal flaw, but one that needs to be discussed openly. Poor stage separation would probably mean that the results in figure 5 are even more striking.

2) The authors should also see Testis FCA, PMID: 36795469 for a good example of how to validate a testes annotation. Lifting over from this outstanding Drosophila effort is a good start, but probably not completely satisfactory, without showing that a significant number of those annotations can be confirmed by in situ or other direct methods in the mosquito (maybe two+ spermatogonia and two+ primary spermatocyte examples). As it stands, I'm not really convinced of the cell types, which lowers confidence in the conclusions.

3) MSCI is visible in many species at the cell biology level. Backing up Figure 5 with data on chromosome condensation, histone marks, or measurements of nascent transcripts would greatly increase my enthusiasm.

4) Figure 6 presents an interesting potential mechanism of reduced upregulation of the X rather than downregulation, which should be discussed more thoroughly in light of papers making the claim that the X is actively inactivated.

5) The references and resulting discussion are quite out-of-date. The authors need to carefully read and discuss the following papers, which are directly related to the X chromosome work in the manuscript and have already discussed many of the points that the authors are making. There are also some newer ones appearing in bioarchives, so a fresh literature search and resulting edits will be needed. Here are a few examples of critical literature for the topic. This will make for a more significant contribution to the literature and will appropriately credit those that have been advancing this field via single cell methods (Fly Cell Atlas, PMID: 35239393; Testis FCA, PMID: 36795469; Testis X chromosome, PMID: 33563972; Mammalian testes, PMID: 36544022; etc.).

Response to Reviewers

Thanks to all the reviewers for the time they have taken and their detailed and constructive suggestions for the manuscript. The manuscript is greatly improved as result of having addressed these. Our responses are detailed below:

Reviewer 1:

*This study by Page et al. uses RNA expression profiling of testes single cells to analyze spermatogenesis cell types in the mosquito *Anopheles gambiae*, which is an important vector of human diseases. One of the main goals of this study is to develop a catalogue of candidate genes with stage-specific expression during spermatogenesis, which can then be used for fine-tuned activation of transgenes for various vector control strategies. Another aim of this study is to identify the precise stage at which meiotic sex chromosome inactivation (MSCI) is initiated. In addition to the general interest in the mechanisms of MSCI, understanding the stage at which it starts can facilitate candidate selection to avoid issues with artificially silencing X-linked genes.*

*The study is executed very well and the analysis clear and succinct. I think it is very clever to use the gene ontology results on the cell clusters and the expression pattern of *aub* to infer the onset of MSCI, and I think the resource will indeed be valuable for the community. One main suggestion: I thought the authors could have done a bit more with the pseudotime trends to corroborate the enriched cell cluster groupings. For example, identify genes that directly correlate with the pseudotime trajectories in addition to the cluster enrichment and DE between cell clusters. It is not guaranteed that this analysis would yield additional candidates for each cell type, but the authors should at least consider it.*

1.1) We thank the reviewer for their appreciation of our manuscript and its impact. Regarding their suggestion around the correlation between unsupervised pseudotime progression and cluster annotation, we have included in Figure 2 the pseudotime plots with expression profiles of a set of marker genes with known expression in the *Anopheles* germline or orthologs of *Drosophila* germ-cell markers which were previously shown as t-SNE plots in the former Figure 3. We agree with the reviewer that the new figure is more effective in conveying the correlation between the spatiotemporal progression of each germline cluster and the corresponding marker genes.

My other suggestions below are fairly minor.
 - Line 90-91: There's a strange reference error here.

1.2) Corrected

- Figure 3: It would be preferable to plot the normalized log-transformed expression values to highlight abundance patterns more clearly. (Based on the value ranges, I'm assuming these are not log-transformed).

1.3) We decided to display linear expression in in Figure 2 because log-transformed values in some cases appeared to graphically mask the peaks in expression of highly expressed genes. Nonetheless, we have added violin plots for the genes used for cluster annotation, showing log transformed expression as supplementary Figure S3.

- Figure 5: legend typo "...of from..."

- Line 234-241: formatting issue.

1.4) Corrected

Reviewer 2:

Summary: Page et al. present work to analyze the transcriptome of Anopheles gambiae testes. This organ has been previously studied by the authors at the cell compartment level using FACS using these same two transgenic reporters followed by high-throughput RNAseq sequencing. Their analysis has been repeated in more detail by single cell RNAseq. While incremental, this technology is better suited to describe the progression of spermatogenesis and capture lowly represented cell populations. While X-linked silencing was known, the authors identify overexpression of X-linked genes in germline stem cells, a tiny fraction of cells. As well as biological interest, this analysis may be of benefit to the wider community to identify regulatory elements for transgene expression that are specific for certain stages of spermatogenesis, although the authors do not validate any new regulatory elements, nor expression patterns by in situ hybridization.

Overall comments:

The data are convincing, and the paper is well written. I have mostly minor comments for clarity. The bioinformatics analysis is interesting and relevant to identifying the onset of germline silencing. There seems to be no reporting of results on the other sex chromosome, the Y chromosome, despite mapping to Y contigs. Is this to be published elsewhere? If not, some comments on the findings would be appropriate e.g. Is Y chromosome also silenced during meiosis and post-meiosis? Are Y-bearing haploid cells transcriptionally different to X-bearing haploid cells?

We thank the reviewer for their positive appraisal of our work and its significance.

2.0 For the purposes of completeness we have now added Y contigs to the analysis done in new Figures 4 and 5. Low expression of genes mapping to the Y contigs, in addition to the repetitive nature of the Y, limited the amount of additional analysis that could usefully be performed for this set of genes in the rest of this paper.

The power of scRNAseq is in its ability to examine heterogeneity in cell populations but the authors do not specifically examine the most/least variable genes in each cluster. This would seem like a missed opportunity given this dataset.

2.1) We have now included a set of genes in new figure (Figure S4) showing the most significantly enriched genes for each cell cluster and provide a full list of differentially expressed genes as Supplementary TableS1.

Detailed comments:

Abstract

No comments.

Introduction

'...transgene expression being insufficiently restricted to the germline. This is thought to be caused by leaky expression in the somatic cells and/or germline-expressed proteins being deposited in the zygote post-fertilization.'

This point seems to be applicable to other strategies as well, not just those where sperm transfers proteins into the zygote but also maternal deposition into the egg is just as relevant. Since eggs and sperm are within the germline, I would say this is a protein/transcript stability issue rather than a lack of restriction to the germline.

2.2) Taking on board the reviewer's comment, we have modified the text to make this clearer as follows:

"One major limitation of these strategies thus far has been the reduced efficiency caused by transgenic nuclease activity being insufficiently restricted to the male or female germline. This is thought to be caused by leaky expression in the somatic cells and/or some germline-expressed transcripts or proteins being deposited into the zygote post-fertilisation."

'Dosage compensation is a widely described cellular mechanisms that equalizes the expression of X-linked genes in organisms containing a sex-determined number of X chromosomes.' This sentence is badly worded in my view and should be revised as it is a very broad statement and the organism is not specified. Sex is not necessarily determined by the number of X chromosomes but by the ratio of X to Autosomes or the presence of a Y chromosome e.g. XY and XXY are both human males.

2.3) We were not trying to say that sex was determined only by the number of X chromosomes but we agree with the reviewer that the sentence was not particularly clear and have now changed it with the following:

*"During spermatogenesis, regulation of gene expression differs between the sex chromosomes and the autosomal chromosomes, primarily due to two mechanisms known as Dosage Compensation and Meiotic Sex chromosome Inactivation. Dosage compensation is a widely described cellular mechanism that balances the expression of sex-linked and autosomal genes in the heterogametic sex such as *A. gambiae* male mosquitoes, which carry only one copy of the X chromosome in each diploid cell, whilst females carry two X chromosomes."*

Results and Discussion:

Isolation of single germline cells from the Anopheles male gonads for single-cell RNA sequencing

Additional details of the scRNAseq results would be helpful if presented, particularly relating to cell quality control, for example, cut-off values for reads per cell, total number of reads, mean reads per cell, % mapping to genome.

2.4) We have now included this information to the Results section as follows:

"Single cell 3' RNA sequencing with 10x genomics provided transcriptomic data for 4443 individual cells a median of 2849 genes expressed per cell and a mean of 56,907 reads per cell. 88.7% of reads mapped to the genome with 65.2% mapped confidently to exonic regions. Cells with a minimum expression of 200 genes expressed were included for downstream analysis."

Typo: ... trajectory analysis (REF error). was performed...

2.5) Corrected

Figure 1b is not referred to in the main text

2.6) Figure 1a and 1b switched and both now properly referenced in text.

The clusters described in Figure 3 and the text differ to those represented in Figure 1. The same region of the t-SNE plot is labelled as Cluster 4 (same color as primary spermatocytes) in Figure 1 and as Spermatogonia / Primary spermatocytes in Figure 3 and I find this discrepancy confusing.

2.7) We have updated the figures mentioned with the new Figure 2a where we have replaced cluster numbers with cell types. We were unable to assign exact colours as some clusters appear to span the boundary between two cell-types.

Why haven't spermatogonia been distinguished from primary spermatocytes? Consider adding some blue color to the primary spermatocytes region in Figure 1b.

2.8) Increasing cluster numbers did not separate these two clusters but instead created increasing sub populations from the mature spermatozoa. It seems that the late spermatogonia and primary spermatocytes have very similar gene expression which is not unusual considering this is largely a growth phase.

Dnah3 doesn't appear to be particularly enriched in spermatozoa in the t-SNE plot as many cells do not express it. Are there other genes that might be more convincing?

2.9) Similarly to previous works in other organisms including *Drosophila* (e.g., <https://doi.org/10.7554/eLife.47138>), our single-cell dataset evidences low level of expression in the latest stages of sperm formation with the lowest RNA content found in the spermatozoa cluster (see Fig 4). This is consistent with transcriptional inactivity of mature spermatozoa, evidenced in other organisms including humans (doi.org/10.1111/j.1439-0272.2005.00656.x). The AGAP007675 gene, ortholog of the *Drosophila* axonemal dynein protein *Dnah3*, was selected as showing significant enrichment and an overall trend (see Fig. 2l & Table S1) consistent with its requirement for sperm motility and trend of expression shown previously in *Anopheles* testes (doi.org/10.1038/s41598-019-51181-1).

How does expression of the fluorescent marker proteins compare with the endogenous gene expression of vasa and B2 tubulin?

2.10) The chemistry used for library preparation (using the 10x Genomics Chromium Single Cell 3' v2 Reagent Kit) allows coverage of the 3' end of genes. This means that it is virtually impossible to discriminate the bulk of reads mapping the 3'UTR of the transgene and the corresponding endogenous sequences making an eventual comparison ineffective.

Differential expression analysis highlights genes with predicted functions compatible with apparent spermatogenic stages – Typo in title?

2.11) The title of this section is changed to: “Cell cluster enriched genes show predicted function compatible with spermatogenic stages.”

In the cases of cluster 1 and 2, and 5 and 8, there are large differences in the number of cells being tested against ‘the rest’. Are the gene enrichment results similar if fewer cells are randomly sub-sampled?

2.12) The number of cells found in each cluster largely reflect the proportion of cells expected throughout spermatogenesis with the caveat determined by our artificial counterselection of mature sperm cells via Fluorescence Activated Cell Sorting as described in the manuscript. We have now included expression profiles of the most significantly enriched genes in each cluster in the Supplementary Table S1 and Figure S4.

Are adjusted p-values presented in Table S1? What is the correction for multiple testing? Consider adding a statistical section to the methods.

2.13) Table S1 contains p-values calculated using the Benjamini-Hochberg correction for multiple testing. Statistics added to methods for clarity.

Typo:...such as RNA processing, ...

2.14) Corrected

Chromosome-wide expression analysis indicated overexpression of the X-chromosome in GSCs and the onset of MSCI.

It's kinda odd to derive a ratio of 1 from median expression values of 0/0 for Hub cells when Figure 5 shows RNA content derived from both chromosome locations. What about if you restrict to genes detected in the cluster? ‘This suggests that additional silencing...’ Why is this additional?

2.15) We corrected the 0/0 ratio to N/A and we corrected the typo “additional silencing” with “silencing”

There is a lack of dosage compensation and a lack of silencing in primary spermatogonia. Table 1: To claim silencing, it would seem appropriate to also test for deviation from a ratio of 0.5.

2,16) X/A ratios used to evaluate eventual presence or absence of dosage compensation are generally calculated using median values. However, previous studies including our own (doi.org/10.1038/s41598-019-51181-1), show that X/A ratios are affected by the threshold of minimum expression applied. For this reason, here we have applied standard threshold and measured how each X:A ratio significantly differs from 1 (null hypothesis) rather than deviation from 0.5, which we know is susceptible to the threshold of expression applied.

Typos:

... shows an apparent increase ...

... distribution of chromosomal location. This ...

Conclusions

Typo:

... an overexpression of X-linked genes...

2.17) Corrected

Methods

How many biological replicates were done?

2.18) Following procedures described in similar peer reviewed studies in other organisms, we have dissected about 100 mosquitoes resulting in 191 testes that were subjected to the single-cell RNA sequencing procedure as unique sample and outputting data from a total of 4443 independent cells. This information is provided in the methods.

Please specify the repository where this data will be publicly available.

2.19) We have now deposited the original data and included the following text to the manuscript:

“The original sequencing data for this paper is available online under accession number PRJNA971569. Any additional data generated during analysis can be made available upon request.”

Figures

Figure 1b is not referred to in the main text.

Some panels of Figure 3 are not referred to in the main text.

Figure 5 title has a typo: ... content of from autosomal...

Figure 6: 1-star significance is not defined in the legend. Wilcoxon should have uppercase W

The headings Client text box are not very intuitive in Table S2

2.20) Corrected

Reviewer 3:

What happens to sex chromosome expression during spermatogenesis and the prospects of using these properties to engineer vector control lines is topical and interesting. The title of the manuscript conveys a deep analysis of the topic. We see a hint of this in the very interesting figure 5. Unfortunately, the manuscript is thin, probably premature, and does not really deliver strong conclusions. It will take some work, but I hope the authors put in the time rather than just turning around the manuscript, as the importance of a convincing work would be very high.

1) While annotating cell types is a long a tedious task, it is of critical importance. The literature is beginning to be polluted with poor annotations, that are then propagated. I find many of the elements of fig 3 worrisome, for example, the authors show genes that I would expect to see late, e.g. protamines and beta tubulin, expressed in the spermatogonia. The hub annotation is not at all convincing. Some of this might be due to the graded scale, which made it hard for me to distinguish high and low expression – so I might be overly harsh as a result. I think the authors should reexamine the clustering granularity and try to see if other resolutions and parameter adjustments could help make the stage separations better. I am not asking for distinct clusters, as this is a continuum, but I suspect this could be improved. If not, I don't think it's a fatal flaw, but one that needs to be discussed openly. Poor stage separation would probably mean that the results in figure 5 are even more striking.

We thank the reviewer for recognising the potential importance of the work and for their constructive suggestions, which we respond to below:

3.1) Cluster annotations

We agree with the reviewer that a comprehensive annotation of cell clusters is important and often essential to obtain or support downstream analysis and conclusions, such as those we were able to extrapolate from our datasets on Dosage Compensation and MSCI. However, annotations rely on availability of truthful cell-type specific “marker” genes or the characterisation of new ones and we are aware that there are multiple complications when applying either of these options in *Anopheles*, particularly to the germline tissues. *Anopheles* gene annotations are still largely relying on orthology inference from the *Drosophila* model, hence our difficulties in providing a more comprehensive set of “marker genes” for our initial annotation of germline clusters, particularly for relatively small and specialised clusters such as the hub. On the other hand, the characterisation of new potential marker genes requires gene specific probes (to detect either their transcripts or protein products), which we know it is difficult to achieve with adequate specificity and resolution that would be necessary to define a minimum of 8 germline-specific clusters in *Anopheles* testes.

Cluster granularity

In our study we have used k-means clustering to group cells with similar patterns of expression and pseudotime trajectory analysis to reconstruct the spatiotemporal progression of germline formation in an unsupervised manner. We have triaged this by inputting different numbers of clusters to be generated and we found that 8 was the maximum number of clusters that could be ordered in a meaningful pseudotime trajectory and annotated using available germline marker genes.

Hub annotation

As we mention in the text, the pre-sorting of cells using red and green fluorescent markers (expressed under distinct germline promoters) should have excluded the majority of somatic cells and possibly most of the hub cells. However, we were able to identify a very small number of cells (11 in total), which showed very low levels of transcript from the markers, and a significant enrichment for AGAP029564 (Log2FC: 9.98 , $p < 0.0001$) ortholog of the *Drosophila* hub-cell marker *Fas3*. We agree with the reviewer that this was not particularly clear from the t-SNA plot, also due to the graded scale used previously, and we have now included both a

UMAP reduction plot and a violin plot (new Fig. 2 and Fig.S3) which indicate a near to exclusive expression of the gene in the hub cluster. In the new version of the manuscript, we have also included additional genes that we have found being enriched in each cluster and, although for the majority of these we do not have a predicted function, one of the top enriched genes we have found in the hub cluster is AGAP000472, which is the ortholog of *Drosophila* short gastrulation gene (*sog*), thought to be produced by hub and cyst cells and involved in stem-cell niche maintenance.

Protamine and beta2 tubulin expression in spermatogonia

We agree with the reviewer that these genes are not expected to show high levels of expression in premeiotic stages. However, we have seen similar unexpected patterns in our previous dataset where we detected relatively high levels of *beta2 tubulin* transcripts in the premeiotic stage (see doi.org/10.1038/s41598-019-51181-1, Supp. Fig. S3). This pattern seems recurrent in highly expressed genes, possibly indicating some level of background mRNA that may be derived from ruptured cells. However, we cannot exclude mechanisms of posttranscriptional regulations that would imply transcripts being produced earlier than expected.

2) The authors should also see Testis FCA, PMID: 36795469 for a good example of how to validate a testes annotation. Lifting over from this outstanding Drosophila effort is a good start, but probably not completely satisfactory, without showing that a significant number of those annotations can be confirmed by in situ or other direct methods in the mosquito (maybe two+ spermatogonia and two+ primary spermatocyte examples). As it stands, I'm not really convinced of the cell types, which lowers confidence in the conclusions.

3.2) We agree with the reviewer that In-Situ Hybridisation or similar approaches would be useful, particularly to fully characterise and/or annotate new cell-type specific marker genes. However, we believe that this goes beyond the direct scope of this work, which is to provide the community with a novel dataset to explore the malaria mosquito spermatogenesis transcriptome with unprecedented resolution. We believe that we provide enough evidence as support of our conclusion and that an eventual validation of a couple of new markers showing cluster-specific transcript would bring a marginal contribute to our conclusions.

3) MSCI is visible in many species at the cell biology level. Backing up Figure 5 with data on chromosome condensation, histone marks, or measurements of nascent transcripts would greatly increase my enthusiasm.

3.3) We agree, and we are also very eager to unveil further mechanisms underlining MSCI in mosquitoes, which again require efforts that are beyond our current capacity and primary research priorities. Nonetheless we believe that our results provide some fundamental transcriptomic evidence regarding this mechanism that was virtually uncharacterised in mosquitoes up to now. The additional experiments suggested would be beyond the scope of this article.

4) Figure 6 presents an interesting potential mechanism of reduced upregulation of the X rather than downregulation, which should be discussed more thoroughly in light of papers making the claim that the X is actively inactivated.

3.4) The X-chromosome “downregulation” we detected here as well as in our previous 4-population dataset (doi.org/10.1038/s41598-019-51181-1) evidence MSCI inactivation, which therefore acts only from spermatocyte onwards and therefore it is not expected to have implications with the possible mechanisms of X upregulation, which we have also detected in our previous data mentioned above. It is difficult for us to establish strong conclusions without further evidence, also because only a very few of the currently annotated X-linked genes show testis expression, apparently associated to a process of “demasculinization of the X chromosome” previously described also in *Anopheles* (doi.org/10.1186/1471-2148-12-69). Although we find this transcriptional signature consistent with our previous findings, we are unable to establish whether this may be associated to the few genes we detect, e.g. as a consequence of cis-regulatory evolution mechanisms as described previously in *Drosophila*, or due to other genetic or epigenetic mechanisms affecting larger regions of the X-chromosome that may be yet poorly annotated.

5) *The references and resulting discussion are quite out-of-date. The authors need to carefully read and discuss the following papers, which are directly related to the X chromosome work in the manuscript and have already discussed many of the points that the authors are making. There are also some newer ones appearing in bioarchives, so a fresh literature search and resulting edits will be needed. Here are a few examples of critical literature for the topic. This will make for a more significant contribution to the literature and will appropriately credit those that have been advancing this field via single cell methods (Fly Cell Atlas, PMID: 35239393; Testis FCA, PMID: 36795469; Testis X chromosome, PMID: 33563972; Mammalian testes, PMID: 36544022; etc.).*

3.5) We thank the reviewer for the suggestions. We have now updated our discussion and included relevant references as follows:

“Similar studies in *Drosophila* also observed reduction in X chromosome expression initiating in the spermatocytes^{26,60}”

Reviewers' comments:

Reviewer #1 (Remarks to the Author):

The revised manuscript adequately addresses my initial comment, and I believe is significantly improved. I have no further concerns.

Reviewer #2 (Remarks to the Author):

The authors have satisfied my concerns and are commended on an improved manuscript.

Reviewer #3 (Remarks to the Author):

While it might take some work, like in situ, antibody staining, and minimally a fuller exploration of where key Dmel testis genes are expressed in the scRNAseq data set, it is essential if the authors want to make claims like the below in the abstract:

"We used single-cell RNA sequencing to further discriminate these populations and define distinct germline cell-types. In doing so, we revealed an overexpression of the X chromosome in the germline stem cells (GSCs) and were able to pinpoint the onset of meiotic silencing of the X chromosome in the late spermatogonia/primary spermatocytes."

Response to Reviewers

We thank all the reviewers for the constructive inputs, which allowed us to further improve clarity and quality of the manuscripts. We have now included some additional information and modified some of the relevant text to clarify the remaining points highlighted by reviewers and summarised below.

Reviewers' comments:

Reviewer #1 (Remarks to the Author):

The revised manuscript adequately addresses my initial comment, and I believe is significantly improved. I have no further concerns.

Reviewer #2 (Remarks to the Author):

The authors have satisfied my concerns and are commended on an improved manuscript. We thank the reviewer for their suggestions and appreciation of our revised manuscript.

Reviewer #3 (Remarks to the Author):

While it might take some work, like in situ, antibody staining, and minimally a fuller exploration of where key Dmel testis genes are expressed in the scRNAseq data set, it is essential if the authors want to make claims like the below in the abstract:

"We used single-cell RNA sequencing to further discriminate these populations and define distinct germline cell-types. In doing so, we revealed an overexpression of the X chromosome in the germline stem cells (GSCs) and were able to pinpoint the onset of meiotic silencing of the X chromosome in the late spermatogonia/primary spermatocytes."

We thank the reviewer for the thorough examination of our manuscript and the constructive suggestions. While we accept that a comprehensive reconstruction and rigorous identification of all distinct cell types would require a larger repertoire of cell type specific 'marker' genes that, as we explained in more details in our previous response, would require additional experimental effort that goes beyond the scope of this work. As suggested by the reviewer, we have modified relevant parts of the text, including abstract and conclusion, to make this limitation clearer. Nonetheless, the mechanisms of gene regulation highlighted through our single-cell analysis (MSCI and Dosage Compensation), and their relative timings through spermatogenesis, are supported by the co-expression of genes with predicted function consistent with the mechanisms highlighted. In line with the reviewer's suggestions, we have now included additional genes that further support these mechanisms acting in distinct clusters.